# Machine Learning-Based Peripheral Artery Disease Identification Using Laboratory-Based Gait Data

**DOI:** 10.3390/s22197432

**Published:** 2022-09-30

**Authors:** Ali Al-Ramini, Mahdi Hassan, Farahnaz Fallahtafti, Mohammad Ali Takallou, Hafizur Rahman, Basheer Qolomany, Iraklis I. Pipinos, Fadi Alsaleem, Sara A. Myers

**Affiliations:** 1Mechanical Engineering Department, University of Nebraska-Lincoln, Lincoln, NE 68588, USA; 2Department of Biomechanics, University of Nebraska at Omaha, Omaha, NE 6160, USA; 3Department of Surgery and VA Research Service, VA Nebraska-Western Iowa Health Care System, Omaha, NE 68105, USA; 4Durham School of Architectural Engineering and Construction, University of Nebraska–Lincoln, Omaha, NE 68182, USA; 5Cyber Systems Department, University of Nebraska at Kearney, Kearney, NE 68849, USA; 6Department of Surgery, University of Nebraska Medical Center, Omaha, NE 68105, USA

**Keywords:** peripheral artery disease, vascular disease, machine learning, gait analysis, deep learning

## Abstract

Peripheral artery disease (PAD) manifests from atherosclerosis, which limits blood flow to the legs and causes changes in muscle structure and function, and in gait performance. PAD is underdiagnosed, which delays treatment and worsens clinical outcomes. To overcome this challenge, the purpose of this study is to develop machine learning (ML) models that distinguish individuals with and without PAD. This is the first step to using ML to identify those with PAD risk early. We built ML models based on previously acquired overground walking biomechanics data from patients with PAD and healthy controls. Gait signatures were characterized using ankle, knee, and hip joint angles, torques, and powers, as well as ground reaction forces (GRF). ML was able to classify those with and without PAD using Neural Networks or Random Forest algorithms with 89% accuracy (0.64 Matthew’s Correlation Coefficient) using all laboratory-based gait variables. Moreover, models using only GRF variables provided up to 87% accuracy (0.64 Matthew’s Correlation Coefficient). These results indicate that ML models can classify those with and without PAD using gait signatures with acceptable performance. Results also show that an ML gait signature model that uses GRF features delivers the most informative data for PAD classification.

## 1. Introduction

Peripheral artery disease (PAD) is a cardiovascular disease caused by atherosclerosis, which limits blood flow to the arteries and tissues. PAD affects up to 10% of Americans over 40 years of age [1,2,3,4]. The number of patients with PAD has increased, making PAD the third most common atherosclerotic cardiovascular disease after coronary artery disease and stroke [5,6,7,8,9]. The most prevalent symptom of PAD is intermittent claudication, defined as ischemic pain that develops when working leg muscles do not have adequate oxygen [10]. Patients with PAD become progressively more sedentary [11,12] and have altered mobility [13,14,15,16,17,18,19,20]. Moreover, functional impairment frequently occurs before PAD diagnosis, and unidentified, asymptomatic PAD is associated with more adverse outcomes than intermittent claudication [21].

Diagnosing PAD early would enable treatment to slow disease progression, which would decrease the risk of major cardiovascular events [21]. However, 40–60% of patients with PAD go undiagnosed in a primary care setting [21]. The standard diagnostic method, the ankle-brachial index, is a highly specialized test that is costly and requires technologists with training in a specialized vascular lab setting [12,22,23,24]. Sheng et al. reported that pulse wave measurements could accurately detect PAD as ABI, but the pulse wave recording technique could be affected by physiological limitations [25]. A pulse wave depends on peripheral blood flow and may be affected by sympathetic nerve input rather than vessel patency [25]. In addition, severe congestive heart failure can also simulate inflow disease by reducing the blood flow [25]. A review study suggested that although pulse wave velocity measurements to detect PAD are reliable hemodynamic measures, further research is needed to establish the screening and diagnostic validity [26]. The diagnosis of PAD is challenging because of the absence of a distinctive sign that can help physicians to distinguish PAD from the typical signs of aging and other movement-related health conditions. A non-invasive screening approach that physicians could use to identify individuals at higher risk of PAD during daily activities is needed.

Recent research has implemented a data-driven approach using machine learning (ML) to identify patients with PAD [27,28,29,30]. ML models have been implemented for PAD diagnosis using blood samples and Doppler data [16]; clinical records [17]; symptom surveys, interviews, and walking distances [18]; and arterial pulse waveforms [19]. While some of these diagnostic models achieved accuracies up to 87%, significant limitations arise in terms of time required (e.g., multiple years of medical records), resources (e.g., protein-based lab setting and interviews that are not standard of care), and involvement of experts with advanced training to obtain the required data to train the models. A model using the six-minute walk test and symptom scores has fewer barriers but led to a compromised accuracy of 69%, and it still required detailed physician evaluation to gather symptom scores [18]. ML and Neural Networks have also been used to automate the classification of arterial segments affected following PAD diagnosis. This approach used computer vision algorithms with Doppler waveforms and PAD imaging but also required manual adjustment of images, which is time consuming [31,32]. Deep learning-based arterial pulse waveform analysis was also used to detect and estimate PAD severity, but this test is not easier to access than an ankle-brachial index test [30]. Other research developed ML algorithms to identify PAD and predict the mortality risk using complete clinical, imaging, demographic, and genomic information for each patient [28]. The resulting machine-learned models surpassed stepwise logistic regression models to identify patients with PAD and predict future mortality. However, the models depended on the availability of multiple clinical information collected simultaneously, which may not be available in practice and would only be helpful after a PAD diagnosis [28].

ML has recently been applied to ground reaction forces and joint angles to characterize gait in individuals [33], including those with Parkinson’s disease [34,35,36], but not for PAD. Gait analysis has proven crucial in determining the mechanisms and severity of functional limitations, measuring treatment effectiveness, and monitoring the progression of PAD [18,37,38]. For instance, patients with PAD walk slower, have decreased step length while walking before and after pain onset, and spend more time in the double support phase compared to older healthy controls [18,19,20,39]. In addition, gait biomechanics studies have found reduced joint angular displacement, velocities, and accelerations in patients with PAD compared to older controls without PAD [40]. Based on the consistently altered gait patterns in patients with PAD [18,19,20,38,41,42,43,44], it is likely that ML can be applied to gait data to identify the presence of PAD. Thus, ML can be valuable for developing gait signatures that enable early PAD detection and monitor functional severity, disease progression, and improvements with treatment.

This paper implements ML models on gait features to distinguish individuals with PAD from healthy older individuals without PAD. The organization of this paper is as follows: First, we provide a detailed description of the data sources, including gait data and the produced gait features. Next, we provide a preliminary descriptive analysis of gait signatures by studying variance, F-statistic, information gain, and correlation among the gait features. Then, we describe the predictive ML models, including data extraction, grouping, and feeding of the ML model. Finally, we dive into our ML approach that uses gait signatures to classify PAD by extracting the most distinguishing gait features. Figure 1 briefly demonstrates this paper’s workflow. This work provides a foundation to model PAD gait features from biomechanics data collected in the lab. This modeling approach may inspire extracting those gait features from acceleration measurements taken with wearable devices, which could be worn in real-world settings to identify potential patients with PAD. Thus, the presented work takes an essential first step toward continuously monitoring individuals’ physical and movement behavior. PAD is costly for individuals, governments, and society. These new models could be used to monitor moving in the real world, helping alert physicians to the potential presence of PAD in general practices, enable in-home detection of worsening PAD symptoms, manage chronic PAD, and predict when significant adverse health events may occur.

## 2. Data Sources

This section describes the available biomechanics data and the gait feature extraction process for ML applications. Biomechanics data were gathered from studies conducted and approved by the Institutional Review Boards at the University of Nebraska Medical Center and the Nebraska-Western Iowa Veteran Affairs Medical Center. These studies consist of a total of 270 participants, including 227 patients with PAD and 43 healthy older controls.

Experimental tests were conducted in the Biomechanics Research Building gait lab at the University of Nebraska at Omaha. Reflective markers were placed at specific anatomical locations on the lower limbs, utilizing the marker systems of Vaughan [45] and Nigg [46]. Each subject walked at their self-selected pace through a ten-meter pathway containing force platforms set level with the floor. Kinematics data were recorded using a 12 high-speed digital camera motion capture system (100 Hz; Cortex 5.1, Motion Analysis Corp., Rohnert Park, CA, USA) and ground reaction forces were collected using force plates (1000 Hz; AMTI, Watertown, MA, USA). Each patient performed the walking test before (pain-free condition) and after the onset of claudication pain (pain condition). Patients were required to rest one minute in between trials to prevent the onset of claudication pain during the pain-free walking condition. Healthy controls only performed the test in the pain-free condition since they do not experience claudication pain. A total of five successful overground walking trials per leg were collected in which heel-strike and toe-off events were within the boundaries of the force plate. Data were exported and processed using custom laboratory codes in MATLAB software (MathWorks Inc., Natick, MA, USA) and Visual 3D software (C-Motion, Inc., Germantown, MD, USA). Visual 3D software was used to calculate the ground reaction forces in vertical, anterior–posterior, and medial–lateral directions, as well as ankle, knee, and hip joint angles, and joint angular velocities during the stance phase of walking. Joint torques and powers were calculated using inverse dynamics for the ankle, knee, and hip joints during the stance phase of walking. Inverse dynamics combines the kinematics and the ground reaction forces described by Winter [47]. The joint torques and powers determine the lower extremity joint angles, muscular responses (torques), and contributions (powers) during walking.

From the biomechanics overground time-series data (Appendix B, Figure A1), peak discrete points were extracted from all trials for all subjects. Points included minimums and maximums for joint angles, torques, and powers for the ankle, knee, and hip. There were peak values from the anterior–posterior, medial–lateral, and vertical ground reaction forces. Overall, this resulted in a total of 31 predictive gait features from each trial. The peak discrete points were averaged across the five trials for each subject (Appendix B, Table A1), describing the gait features we used to develop the ML models.

## 3. Descriptive Data Analysis

For the preliminary data analysis of gait signatures, we first used statistical methods to provide a descriptive analysis of the data and anticipate the significance of each gait feature by exploring the variance [48], ANOVA F-statistic [49], information gain [50], and correlation [51]. These methods help with feature selection, producing features with high predictive capability for our ML model. Additionally, in our analysis, we used these methods to provide insight into the most important gait feature groups for our classification task (separating patients with PAD from healthy controls). For instance, variance is necessary within the dataset because the significant differences within feature variance allow the ML model to learn the different patterns hidden in the data. F-statistic, or F-test, is a statistical test that calculates the ratio between variance values, such as the variance from two different samples. F-statistics compare and identify relevant features for a classification task. Information gain measures the association between inputs and outputs. Thus, the higher the information gained, the better. Moreover, using the correlation in the data to extract the redundant features produces a better prediction [52]. Then, we utilize the insights from the descriptive analysis to build ML models using sub-groups of the data based on the source of the measurement (ankle, hip, knee, and GRF).

Our dataset consists of 32 predictive features that include 31 numerical features, as described in Table A1 in the Appendix B. First, we use variance analysis to visualize the distribution within each numeric feature and its corresponding presence or absence of PAD. The variance provides insight into the ability of each feature to distinguish between individuals with and without PAD. Because some features in the data are not normally distributed, we used Levene’s test [53], which accepts non-normal distributions to detect the features that significantly differed in variance between individuals with and without PAD. Figure 2 shows a boxplot of each numeric feature and distribution for each group. We divided the figures into four groups based on the gait feature measurement source (ankle, hip, knee, and GRF). The green asterisks represent a significant Levene’s test *p*-values for the difference in variance between patients with PAD and healthy controls. Our results show that GRF features have a higher variance than other features. For example, only one of the gait kinematic (ankle, knee, or hip) features differed in variance. However, there was a difference in variance for most GRF features. This suggests that GRF might be more discriminant gait measurement source to identify PAD.

Another commonly used feature selection method for classification is information gain, which measures the reduction in entropy by splitting a dataset according to a given value of a random variable. Entropy quantifies how much information exists in a random variable, specifically its probability distribution. For example, a skewed distribution has low entropy, whereas a distribution where events have equal probability has a higher entropy. The higher the information gain, the more informative the feature for the model classification capability. Similarly, F-statistic calculates the difference between two sample variances, and the higher F-statistic, the more valuable the data feature. Figure 3 shows the aggregated F-statistic and information gain average based on the measurement source. While we applied our analysis to all gait features, we only show the aggregated average groups of joint measurements for better presentation.

The final step before applying ML is determining the correlation between gait features. This can be used to identify and eliminate redundant features. A correlation study between all numeric features shows that GRF features are highly correlated compared to other features (Figure 4), which indicates that only a few GRF features might be sufficient for distinguishing PAD. This finding, along with our observations from the variance analysis (Figure 2) and F-statistic (Figure 3), suggests that a few laboratory-based gait features, e.g., GRF features, can successfully train ML models to identify the presence or absence of PAD. We test this hypothesis in the upcoming ML section.

## 4. Predictive ML Models to Diagnose PAD

This section explains how we distinguish between patients with PAD and healthy controls. It also demonstrates how the most essential gait features were extracted to identify the presence or absence of PAD. The ML model includes input, data grouping, ML training, ML testing, and performance evaluation (Figure 5). The goal is to find the lowest number of gait features that produce the most accurate ML model to classify individuals as having or not having PAD.

We divided the data into six groups to identify the most important features for classifying PAD (Figure 5). The groups range for including all features (Group 1), features from only one joint (Group 2, 3, 4), all GRF features (Group 5), or all features except GRF features (Group 6). Next, we divided the data into training and testing data sets. Due to the imbalance between the number of healthy controls versus patients with PAD, we oversampled.

The healthy subjects’ data in the training set using the Synthetic Minority Oversampling Technique (SMOTE) algorithm [54]. Subsequently, we applied several ML algorithms to Group 1 training data sets, extracted binary predictions using the test data, and compared these predictions with the original test data. Finally, we used the best algorithms obtained from Group 1 data and evaluated these algorithms for the other groups listed above. We followed this grouping criterion to assess the strength of different data sources (ankle, knee, and hip gait features and GRF) in distinguishing PAD using ML models. The descriptive data analysis in the previous section suggests that GRF gait features might be sufficient for a ML model to distinguish between patients with PAD and healthy controls. Moreover, identifying one valuable gait signature source could minimize the time and computational cost of detecting PAD.

In terms of ML, we used four well-known algorithms: Neural Networks [55,56], Random Forest [57], Support Vector Machine (SVM) [58], and Logistic Regression [59]. Previous research used these algorithms in many classification tasks for diagnostic applications in the medical fields [31,32,60,61,62]. We ran each group to achieve the best predictions and find the minimum features that produced acceptable performance. Specifically, we built the ML models to deliver the best possible performance using Group 1 data (all features), which we treated as a benchmark for comparing our predictions using the other groups.

We used TensorFlow to build Neural Networks models [63], and the Sci-kit Learn library [64] was used to create Random Forest, SVM, and logistic regression models in python [65] (The source code for each ML model can be found in the Appendix A). Then, we used the grid search method [66] and manually tuned the hyperparameters to produce the best performance for each model. Hyperparameters refer to parameters within each ML model that require optimization to produce the best possible prediction result. It is noteworthy that Neural Networks models require the assigning of many parameters to form the model architecture, including the number of hidden layers, number of neurons in each hidden layer, activation functions, optimizers, and other hyperparameters. For example, the architecture of our Neural Networks includes activation functions before each hidden layer, five hidden layers, and an output activation function (Sigmoid) for binary classification.

Random Forest, SVM, and logistic regression ML algorithms require a relatively short building and tuning time to produce the best possible model for our specific data combinations. On the other hand, Neural Networks models need more time to build and tune, requiring many hyperparameters for tuning (Table 1). Neural Networks also require longer training time compares to Random Forest, SVM, and logistic regression approaches. Nevertheless, previous research proved Neural Networks to be a valuable and accurate tool in classification tasks [56]. Each algorithm has unique corresponding hyperparameters, and Neural Networks models require consideration of many hyperparameters (Table 1). (The best possible hyperparameters for each model can be found in the Appendix A).

In the current study, 81% of the total data came from patients with PAD. Therefore, we oversampled the healthy subjects’ data and used multiple performance metrics to add to the accuracy metric (the number of predictions correctly predicted divided by the total number of examples) to avoid over-optimistic results [67]. Here, we provide an overview of some of the metrics we use. These include Matthew’s Correlation Coefficient, Discriminant Power, and Geometric Mean [67,68].

Matthew’s Correlation Coefficient is calculated based on four scores (true positive, true negative, false positive, and false negative), known as confusion matrix scores. Matthew’s Correlation Coefficient provides a good score, which usually ranges between 0.5 and 1 only if the model performs well in all four confusion matrix categories, and it is the least influenced model metric by data imbalance. Thus, we use Matthew’s Correlation Coefficient as the primary comparison method between the models. Matthew’s Correlation Coefficient values ranged from −1 to 1, with “1” as the perfect model, “−1” as the worst model, and 0 no better than a random naïve model [67,68]. The Geometric Mean measures the balance of the classification performance in the majority (PAD) and minority (healthy control) classes. It also helps avoid overfitting in negative and underfitting in positive classes. Finally, Discriminant Power measures the ability of the classifier to distinguish between minority and majority cases. A higher Discriminant Power value translates to better model performance.

## 5. ML Models Results

The predictions of Group 1, which included all gait features, yielded higher performance values in all metrics with the Neural Networks and Random Forest than with SVM and logistic regression (Figure 6). Therefore, in the next step, we applied Neural Networks and Random Forest models to the rest of the groups and compared the predictive performances with Group 1 predictions to measure the ability of the models to classify PAD using a few laboratory-based gait features.

In Groups 2, 3, and 4, which included the gait data generated from one joint (ankle, knee, or hip), the model predictions show that Neural Networks models were more accurate than Random Forest (Table 2). However, all these models were less accurate than the GRF model (Group 5). Group 5 GRF data yielded a comparable prediction accuracy to Group 1 (Figure 7). GRF-based Random Forest predictions had higher Discriminant Power and Geometric Mean values than Group 1 predictions (Table 2). Moreover, all the models that included GRF (Groups 1 and 5) performed better than the models built to distinguish PAD with gait data from only one joint. Interestingly, when all the body joints data were combined (Group 6), the GRF models still performed better, indicating that GRF is a crucial measurement factor in classifying PAD.

Finally, we explored using a few GRF-based gait features by dividing GRF data into three sub-groups based on the signal origins (X: anteroposterior component, Y: mediolateral component, Z: vertical component). GRF-anteroposterior produces five gait features, GRF-mediolateral produces two gait features, and GRF-vertical produces three features (Table A1). GRF-anteroposterior performed better than GRF-mediolateral and GRF-vertical in identifying the presence of PAD (Figure 8). However, using all GRF components still provided better predictive results indicating that limiting the GRF components compromised the model’s ability to identify PAD.

## 6. Discussion

We have described a proof-of-concept application of ML to classify individuals as having or not having PAD using laboratory-based gait features. The model used a dataset of 227 previously diagnosed patients with PAD and 43 healthy controls. Our preliminary analysis based on variance, F-statistic, and information gain (Figure 1 and Figure 2) suggested that GRF gait features hold the most valuable information to classify individuals compared to the joint angle features (ankle, knee, and hip).

Our results (Figure 5 and Figure 6; Table 2) suggest that individuals with PAD have distinct gait signatures compared to healthy individuals, and this ML approach may be helpful in the early identification of PAD. In addition, our findings indicate that classification of PAD is possible using a few laboratory-based gait features. For instance, the Random Forest model based on Group 5 GRF data (Matthew’s Correlation Coefficient: 0.64) performed similar to the model using all gait biomechanics features available in Group 1. Additionally, the Neural Networks GRF-based model (Group 5; Matthew’s Correlation Coefficient: 0.57) yielded comparable prediction quality compared to Group 1 (Matthew’s Correlation Coefficient: 0.64). On the other hand, joint data such as the ankle, knee, or hip produced less accurate results than Group 1. Future research can utilize this information to build a model that monitors disease severity and progression.

Furthermore, we showed the best ML model to handle the predictions for each group. Models using Neural Networks and Random Forest classified individuals with and without PAD using all gait features. The model using the Neural Networks approach produced the highest performance values when using all gait features, while the Random Forest-based model generated the best result for PAD classification using GRF gait features. Finally, we tested the ability of a few GRF laboratory gait features to classify PAD. While some GRF features performed better than the ankle, hip, or knee data alone, the models still lost essential information to provide high-quality predictions, making the model that included all gait features the best.

Based on gait signatures, this study provided a preliminary step toward a more robust model with a larger dataset that can accurately identify the presence of PAD. Moreover, researchers can use this ML method to identify individuals at risk of developing PAD or other diseases using the same training, testing, model architecture, comparison, and performance measurement techniques discussed in this paper. For instance, this research lays the first stone to possibly extract meaningful data points from gait measures captured with wearable sensors by identifying the most crucial gait features. By identifying these features, this research successfully minimizes the complexity of building of ML models that can identify PAD with gait data with larger datasets in the future. Eventually, such models could be implemented based on wearable, real-world data, providing alerts to physicians to order more detailed PAD screening.

There are some limitations to this study. First, the dataset is relatively small for ML applications, so the results from this paper cannot be generalized to identify PAD from a more extensive dataset. In addition, given the challenges of identifying and diagnosing PAD and its severity, the patients with PAD in this study are all pre-labeled with PAD, ensuring that PAD was the primary cause of functional impairment rather than other conditions impact function. Therefore, this could make the results less generalizable. However, this paper demonstrates that ML can offer a high-quality prediction while distinguishing between patients with PAD and healthy controls, even with a few laboratory-based gait features.

Future work will explore the ability of machine learning models to identify early PAD risk and monitor PAD progression and treatment effectiveness. Knowledge from this work could be transferred to wearable sensors that could be integrated into shoes or other assistive devices worn by older individuals. The ability to detect abnormal gait signatures or changes that indicate worsening disease progression can become a valuable tool for managing this chronic disease.

## 7. Conclusions

This paper utilized ML applications to classify individuals with PAD by developing gait signatures with laboratory-based gait features. First, we provided a preliminary analysis to statistically distinguish the essential gait features. Then, we used an ML approach to extract the most valuable features for classifying PAD. Our data-driven approach provided a preliminary foundation for ML identification tasks for an underdiagnosed disease and would greatly benefit from earlier detection. Our findings showed that ML algorithms could produce informative and strong performance values when applied to identify PAD. We also demonstrated that GRF measurements provided better information for classifying individuals with and without PAD. Future research should explore ML models’ ability to identify the risk of having PAD calculating surrogate measures of GRF and gait from measurement taken outside the laboratory. Future work could explore using ML models of wearable gait data as an indicator of PAD risk, as well as to monitor disease progression, severity, and treatment effectiveness.

## Figures and Tables

**Figure 1 sensors-22-07432-f001:**
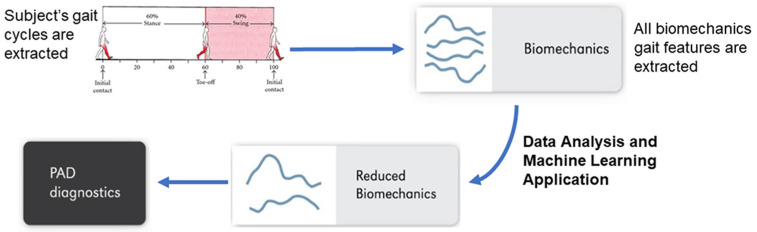
A flowchart briefly describing the utilized data and the methods applied.

**Figure 2 sensors-22-07432-f002:**
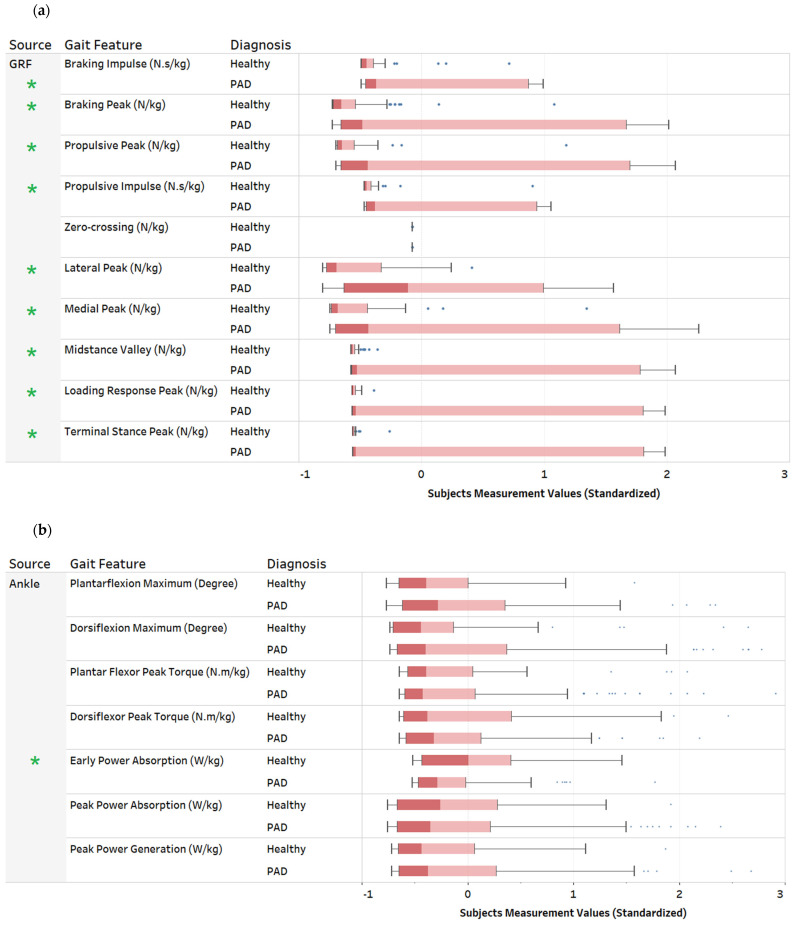
Feature variance comparison of gait features between patients with PAD and healthy controls. (**a**) GRF gait features, (**b**) Ankle gait features, (**c**) Hip gait features, (**d**) Knee gait features. The green asterisks indicate a significant difference in variance based on Levene’s homogeneity of variance.

**Figure 3 sensors-22-07432-f003:**
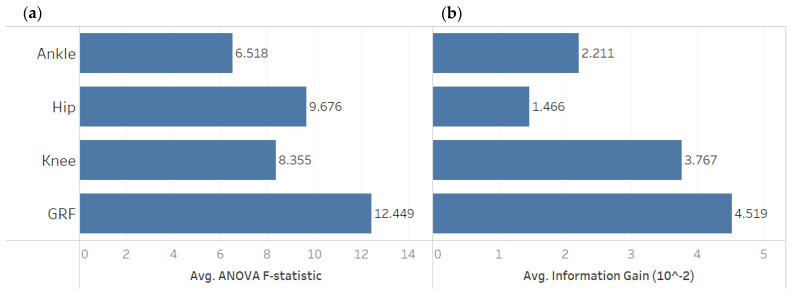
A comparison between gait feature sources in terms of (**a**) Average ANOVA F-statistic and (**b**) Average Information Gain. GRF features have higher F-statistic and information gain than the ankle, knee, and hip gait parameters.

**Figure 4 sensors-22-07432-f004:**
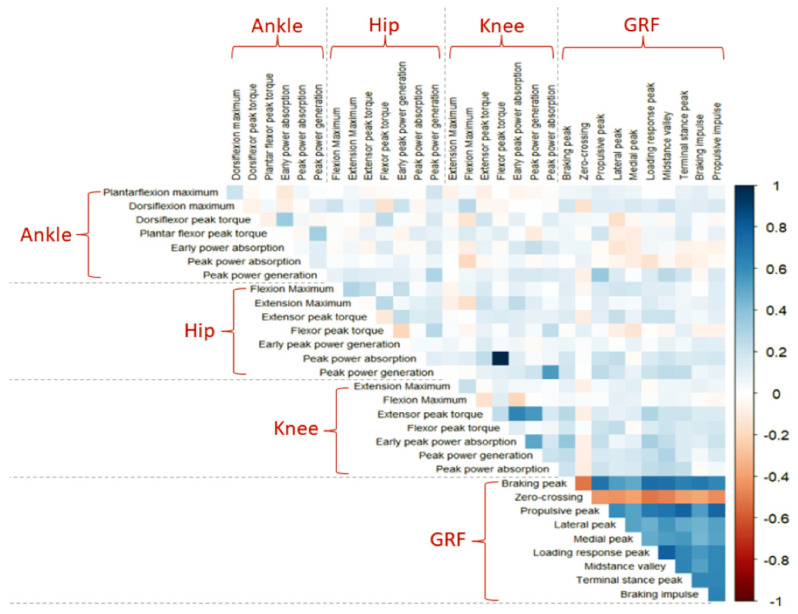
A correlation study of all numeric gait features regardless of PAD status (healthy controls or patients with PAD). The color line indicates the correlation coefficient between features from “Dark Red: −1” to “Navy: 1”. A correlation coefficient of “−1” between two variables implies a perfect negative relationship, and a correlation coefficient of “1” between two variables implies a perfect positive relationship. If the correlation between two variables is 0, there is no linear relationship.

**Figure 5 sensors-22-07432-f005:**
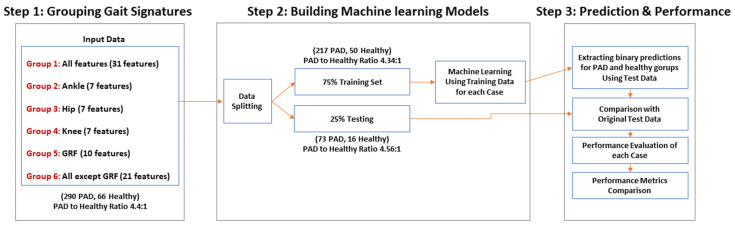
Flowchart showing the 3-step ML analysis method to identify PAD and extract the most valuable gait signatures for PAD diagnosis.

**Figure 6 sensors-22-07432-f006:**
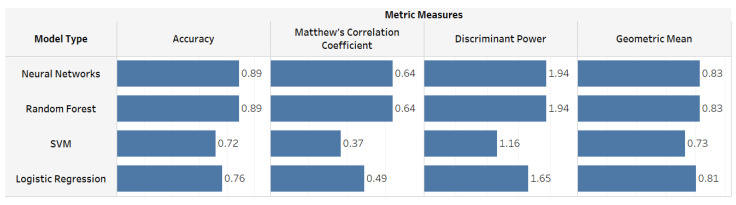
Performance metric results that measure the ability of each ML model algorithm to distinguish PAD.

**Figure 7 sensors-22-07432-f007:**
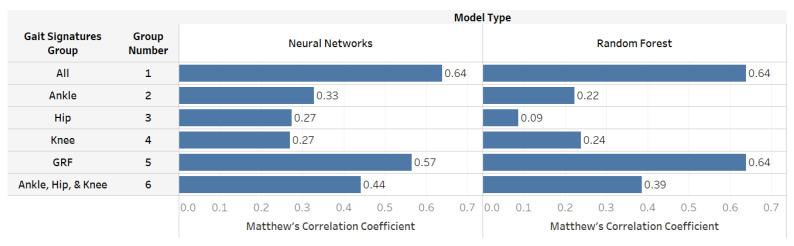
A measurement of the ability of a few laboratory-based gait features to distinguish PAD using Matthew’s Correlation Coefficient. Generally, the GRF-based models (Groups 1 and 5) performed better than joint data models (Groups 2, 3, 4, and 6) and provided comparable results to using all gait features.

**Figure 8 sensors-22-07432-f008:**
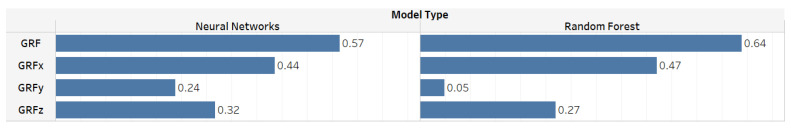
A comparison between GRF-based signals in identifying PAD using Matthew’s Correlation Coefficient as a performance metric. For both Neural Network and Random Forest model approaches, including all GRF components led to better PAD classification compared with any single GRF component (x = anteroposterior, y = mediolateral, and z = vertical).

**Table 1 sensors-22-07432-t001:** The hyperparameters for each algorithm.

Algorithm	List of Hyperparameters
Neural Networks	Activation function Optimizer Kernel Initializer Learning rate Regularization Batch size Number of epochs
Random Forest	Number of trees Maximum number of features Maximum depth of layers Criteria
SVM	Kernel Gamma Penalty parameter
Logistic Regression	Regularization

**Table 2 sensors-22-07432-t002:** A summary of the models’ performances on the testing data. It evaluates the applied model scores utilizing every group based on four performance metrics. The shaded columns highlight the comparison between all gait features models (Group 1) and GRF features (Group 6).

Metric	Model Type	Group Category					
		All	Ankle	Hip	Knee	GRF	Ankle, Hip, Knee
		Group 1	Group 2	Group 3	Group 4	Group 5	Group 6
Accuracy	Neural Networks	0.89	0.79	0.78	0.81	0.82	0.84
	Random Forest	0.89	0.69	0.73	0.75	0.87	0.83
Discriminant Power	Neural Networks	1.94	0.95	0.82	0.90	1.87	1.33
	Random Forest	1.94	0.64	0.29	0.71	2.09	1.19
Geometric Mean	Neural Networks	0.83	0.65	0.61	0.54	0.84	0.84
	Random Forest	0.83	0.63	0.46	0.60	0.87	0.63
Matthew’s Correlation Coefficient	Neural Networks	0.64	0.33	0.27	0.27	0.57	0.44
	Random Forest	0.64	0.22	0.09	0.24	0.64	0.39
Best model type		Neural Networks, Random Forest	Neural Networks	Neural Networks	Neural Networks	Random Forest	Neural Networks
ML Performance Metrics Description: Accuracy: the number of correct predictions divided by the total number of examples. Range: (0 to 1), an accuracy value of “1” means the model predicts perfectly with no errors. Discriminant Power: measures the ability of the classifier to distinguish between minority (healthy controls) and majority (Patients with PAD) cases. A higher Discriminant Power value translates to better model performance. Geometric Mean measures the balance of the classification performance in the majority and minority cases. The higher the geometric mean, the better the model performance. Matthew’s Correlation Coefficient provides a good score only if the model performs well in all four confusion matrix categories. Range: (−1 to 1), with “1” as the perfect model, “−1” as the worst model, and 0 no better than a random naïve model.							

## Data Availability

The data that support the findings and results in the paper can be requested from the corresponding author. A material transfer agreement must be executed between the requesting institutions and the University of Nebraska at Omaha and the Omaha Veterans Affairs Medical Center.

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
