# Peer review of "Machine Learning-Based Peripheral Artery Disease Identification Using Laboratory-Based Gait Data"

_sensors, 2022, doi:10.3390/s22197432_

Round 1
Reviewer 1 Report
Overall, it is an interesting task. The task can be further improved by
a- Need to improve abstract, accuracy is just 87%, try to improve and add some new model of deep learning.
b- In introduction section, cite some relevant article.
1) Iqbal, Muhammad Shahid, et al. "Deep learning recognition of diseased and normal cell representation." Transactions on Emerging Telecommunications Technologies (2020): e4017.
c- Data source section, need to be more clear.
d- Figures 1, and 4 text is so small, not read able.
e- Result and discussion needed to be more explaining and clear.
f- Proposed method, full flow diagram, need to be add.
This is an interesting topic and is generally well presented. The English expression, however, does need attention. I suggest that you get assistance with this and resubmit
Author Response
Thank you for your comments. Our responses are in the attached document.

Reviewer 2 Report
Well presented and clear work with a lot of patient trials conducted.
The research motivation stated as "The standard diagnostic method, the ankle-brachial index, is a highly specialized test that is costly and requires technologists with training in a specialized vascular lab setting [12,22–24]" remains unconvincing - differential BP measurement is not necessarily more complicated than image based gait monitoring, cited related papers are relatively old, newer methods (i.e. pulse wave measurements with low cost wearables) should be properly (honestly) evaluated as well.
Paper conclusions: the presence or absence of PAD using GRF assessment method seems is evident in Fig 1 without of use of any ML algorithms. Motivation to include GRF method to the particular ML focused study should be explained more in details. From the other side, pressure sensor mats are specific medical devices but range of motion (ROM) of joints can be measured with simple (lower cost?) wearable devices. Perhaps more analysis to be added in this subtopic.
Author Response

(The authors gave the same response as above.)

Reviewer 3 Report
In this paper, the authors present a machine learning based peripheral artery disease identification mechanism. The paper is well written. However, I have a few comments on the paper which are listed below. The structure of the paper seems rather unconventional.
After the introduction and data sections, the authors go straight to results. There are details about predictive ML model in section 3.2 of the paper. But I would suggest to keep it as a separate section and place it before results.
Based on the discussion presented in the paper, neural network based model produces better results as compared to random forest. But did authors compare the complexity of two algorithms too? Did authors compare the two algorithms in terms of speed and efficiency? That would be an interesting way to look at the results.
As the authors suggested themselves, a dataset of 227 for an ML model is too small to establish the superiority over other ML models. In my opinion, at the end, for a significantly large dataset, it would come down to complexity of models and speed rather than accuracy. Historically, for a significantly large datasets, ML models do not vary too much in terms of accuracy. The authors need to look at the results from that aspect too.
Author Response

(The authors gave the same response as above.)

Round 2
Reviewer 1 Report
Revision have done according to the comments.
Reviewer 3 Report
To a large extent, the authors have addressed the comments raised previously.